# Overexpression of Sugarcane *ScDIR* Genes Enhances Drought Tolerance in *Nicotiana benthamiana*

**DOI:** 10.3390/ijms23105340

**Published:** 2022-05-10

**Authors:** Xiufang Li, Zongling Liu, Haiyun Zhao, Xingli Deng, Yizu Su, Ru Li, Baoshan Chen

**Affiliations:** 1State Key Laboratory for Conservation and Utilization of Subtropical Agro−Bioresources, College of Life Science and Technology, Guangxi University, Nanning 530004, China; 1908301029@st.gxu.edu.cn (X.L.); 1808401012@st.gxu.edu.cn (Z.L.); 2008301015@st.gxu.edu.cn (X.D.); 2008301053@st.gxu.edu.cn (Y.S.); 2Guangxi Key Laboratory of Sugarcane Biology, College of Agriculture, Guangxi University, Nanning 530004, China; 1917304025@st.gxu.edu.cn

**Keywords:** sugarcane, *ScDIR*, drought tolerance, *Nicotiana benthamiana*

## Abstract

Dirigent proteins (DIRs) are known to function in lignin biogenesis and to be involved in stress resistance in plants. However, the sugarcane DIRs have not been functionally characterized. In this study, we investigated the DIR−protein−encoding genes in *Saccharum* spp. (ScDIR) by screening collections of sugarcane databases, monitoring the responses of these genes to drought stress by real−time quantitative PCR, and identifying their heterologous expression in tobacco. Of the 64 *ScDIRs* identified, four belonging to the DIR−b/d (*ScDIR5* and *ScDIR11*) and DIR−c (*ScDIR7* and *ScDIR40*) subfamilies showed a significant transcriptional response when subjected to drought stress. ScDIR5, ScDIR7, and ScDIR11 are localized in the cell membrane, whereas ScDIR40 is found in the cell wall. The overexpression of these *ScDIR* genes in tobacco generally increased the drought tolerance of the transgenic lines, with ScDIR7 conferring the highest degree of drought tolerance. The characterization of the physiological and biochemical indicators (superoxide dismutase, catalase, malondialdehyde, and H_2_O_2_) confirmed that the ScDIR−overexpressing lines outperformed the wild type. These results demonstrated that specific *ScDIRs* in sugarcane respond and contribute to tolerance of drought stress, shedding light on potential means of improving drought tolerance in this crop.

## 1. Introduction

Plant growth and agricultural production are greatly affected by environmental stresses such as drought, extreme climates, and pathogen attacks. The most direct effect of drought on plants is protoplasm dehydration, which can cause various types of damage. For example, the cell membrane loses semi−permeability, photosynthesis decreases, and respiration increases [1,2]. After drought stress, the photosynthetic rate of plants decreases, the superoxide dismutase (SOD) contents increase, oxygen−free radicals are scavenged, and the normal growth state of plants is restored [3].

Since the first Dirigent (DIR) protein was discovered in *Forsythia suspense* [4], DIR and DIR−like proteins have been identified in various plants [5,6,7]. DIR proteins constitute a large family in plants; for example, *Picea sitchensis* has 19 PsDIR proteins [6,7], and *Oryza sativa* has 49 OsDIR proteins [8]. This protein family is characterized by the presence of a structurally conserved DIR domain and a JAC domain. The DIR protein family is currently divided into eight subfamilies, DIR−a, DIR−b/d, DIR−c, DIR−e, DIR−f, DIR−g, DIR−h, and DIR−i [9]. To date, studies on DIR proteins in plants have mainly focused on the DIR−a subfamily, which is involved in plant resistance to pathogen and pest stresses [7,10,11]. Studies have found that *BhDIR1* transcripts accumulate when plants are in a state of dehydration [12]. The AtDIR6 protein of *Arabidopsis thaliana* can mediate the synthesis of lignin monomer pinoresinol, a secondary metabolite, under abiotic stress [13]. The transfer of the *Bryum argenteum* gene *BaDBL1* into *Arabidopsis* increased the lignin content and enhanced the tolerance to osmotic and salt stresses [14].

Sugarcane (*Saccharum officinarum* L.) is a major sucrose and bioenergy crop grown mainly in tropical and subtropical regions. Drought is the main factor affecting the growth and development of sugarcane [15]. However, the mechanisms underlying sugarcane drought resistance are not well understood. In this study, we investigated the DIR−protein−encoding genes in sugarcane by monitoring their responses to drought stress and identifying their heterologous expression in tobacco. We showed that a small number of *ScDIRs* responded to drought stress in sugarcane and that the overexpression of these genes enhanced the drought tolerance of the transgenic tobacco lines. These results enrich our understanding of the role of DIRs in sugarcane and indicate a possible approach to improving the drought tolerance of sugarcane based on *DIRs*.

## 2. Results

### 2.1. Characterization of Sugarcane ScDIR Proteins

By searching the sugarcane genome, sugarcane expressed sequence tag (SUCEST), RiceData, and GenBank databases, we obtained 64 sugarcane DIR protein sequences (ScDIRs) and 40 DIR protein sequences from other plant species, including *A. thaliana*, *F. suspense*, *Gossypium barbadense*, *Hordeum vulgare*, *O. sativa*, *Picea glauca*, *Pisum sativum*, *Podophyllum peltatum*, *Nicotiana tabacum*, *Sorghum bicolor*, *Triticum aestivum*, and *Tsuga heterophylla*. All of these proteins contained a DIR structural domain; a small number also contained Jacalin (JAC) and Protein Kinase (PK) Domain (Appendix A). Phylogenetic analysis revealed that the ScDIRs were distributed among six subfamilies, with 24 ScDIRs in the DIR−c subfamily and 14 in the ScDIR−b/d subfamily (Figure 1).

### 2.2. Expression Pattern of Sugarcane ScDIR Genes under Drought Stress

To determine whether the expression of *ScDIR* genes was responsive to abiotic stresses, the expression patterns of these genes in sugarcane were examined. Sugarcane seedlings showed the appearance of yellowish dwarf plants after 5 days of drought treatment and died after 10 days of treatment (Figure 2a). We measured the expression levels of sugarcane 64 *ScDIR* genes after 5 days of drought stress; 13 *ScDIR* genes had expression levels more than twice as high as before stress. *ScDIR5*, *ScDIR7*, *ScDIR11*, and *ScDIR40* showed the greatest change in expression (Figure 2b).

### 2.3. Genetic Structures, Sequence Alignments, Motifs, and Tertiary Structures of the ScDIR5, ScDIR7, ScDIR11, and ScDIR40 Genes

Based on the protein family classification described above, ScDIR5 and ScDIR11 were found to belong to the DIR−b/d subfamily, whereas ScDIR7 and ScDIR40 were classified in the DIR−c subfamily (Figure 1). Although the sequence identity of the four ScDIR proteins was only 28.64% (Figure 3a), three identical motifs (motif 1, motif 3, and motif 5) were present in the DIR domain of each protein (Figure 3b,c). Protein sequence analysis showed that the *ScDIR40* gene had two typical DIR structural domains, whereas the other three proteins had only one DIR structural domain (Figure 3c).

The predicted tertiary structures of ScDIR5, ScDIR7, and ScDIR11 were highly similar, with a β−barrel consisting of eight antiparallel chains and three β−barrels converging to form a DIR protein with a hydrophobic cavity in the middle. However, two tertiary structures were predicted in ScDIR40; one of them was somewhat similar to those of ScDIR5, ScDIR7, and ScDIR11, whereas the other looked very different (Figure 3d). These two structures might indicate that one of the two DIR domains in ScDIR40 (Figure 3c) is not functional.

### 2.4. Subcellular Localization of ScDIRs

As DIR proteins are mainly involved in the formation of plant cell walls, we assessed the subcellular localization of ScDIR5, ScDIR7, ScDIR11, and ScDIR40 using transient expression in onion epidermal cells. Green fluorescence signals for ScDIR5, ScDIR7, and ScDIR11 proteins were observed in the cell membrane, whereas the fusion protein for the ScDIR40 protein was detected in the cell wall (Figure 4), which was consistent with the predicted results (Appendix A).

### 2.5. Overexpression of ScDIR Genes Enhances Tolerance to Drought Stress in Transgenic Tobacco

To further characterize the roles of *ScDIR5*, *ScDIR7*, *ScDIR11*, and *ScDIR40* in drought resistance, we expressed these genes in tobacco plants. Transgenic lines were selected for analysis (Figure 5). Leaves of the *ScDIR7*−transgenic and *ScDIR11*−transgenic tobacco plants exhibited increases in lignin content of 87–100% compared with the wild type. Although the lignin contents of *ScDIR5*−transgenic and *ScDIR40*−transgenic plants were lower, they were 1.3 times higher than those of wild−type plants (Figure 6). These results establish a positive correlation between *ScDIR* expression and the lignin content of tobacco.

PEG was used to simulate drought stress in transgenic tobacco [16]. After 15 days of drought treatment, the wild−type, empty vector (EV)−transgenic, *ScDIR5*−transgenic, and *ScDIR40*−transgenic lines gradually wilted, whereas the *ScDIR7*−transgenic and *ScDIR11*−transgenic lines grew and developed normally (Figure 7a). These results demonstrate that the overexpression of *ScDIR* can help tobacco survive for a longer period when exposed to drought conditions.

Then, we analyzed physiological and biochemical indicators related to drought stress; the results showed no significant change in the indicators before treatment. However, after drought stress, the activities of SOD and CAT gradually decreased, and MDA and H_2_O_2_ began to accumulate. After 15 days of stress: SOD and CAT activities were extremely low in wild−type plants and EV−transgenic lines, whereas SOD and CAT activities in transgenic lines were 30–40 times higher than those in the wild type (Figure 7b). After 15 days of stress, the MDA and H_2_O_2_ contents of the transgenic plants were less than half of those of the wild type (Figure 7d,e).

After combining the correlation matrix from the drought tolerance test and the coefficients of physiological and biochemical indexes, PCA and comprehensive evaluation were carried out. The results showed that SOD activity was negatively correlated with MDA and H_2_O_2_ content and that CAT activity was positively correlated with SOD activity (Appendix A). The drought tolerance levels of the four genotypes could be ranked (from high to low) as follows: ScDIR7−transgenic, ScDIR11−transgenic, ScDIR40−transgenic, ScDIR5−transgenic, wild type, EV−transgenic (Table 1).

## 3. Discussion

The DIR protein family is involved in a variety of biological functions. Members of this family have been shown to have important roles in plant resistance to pathogenic bacteria and resilience to external environmental factors. For example, transgenic cotton overexpressing the *GhDIR1* gene has increased the tolerance to *Verticillium dahlia* [17]. The North American *Pissodes strobi* attacked the bark of *p. sitchensis*, and the expression of the *PsDIR* gene was significantly increased [6]. However, in sugarcane, when seedlings were stressed with H_2_O_2_, PEG, or NaCl, the expression of *ScDIR* genes was significantly increased [18]. In this study, we report 64 sugarcane *ScDIR* genes, of which four significantly respond to sugarcane drought stress. The expression of four *ScDIR* genes in tobacco, respectively, revealed the important role of *ScDIR* genes in plant drought tolerance, with the *ScDIR7* gene having the most important role. Consistent with Wu’s finding of rapid accumulation of the *BhDIR1* gene in *Boea hygrometrica* following plant dehydration [12], this demonstrates that sugarcane *ScDIR* genes respond to drought stress.

Recently, the DIR protein family has been classified several times and is currently divided into eight subfamilies: DIR−a, DIR−b/d, DIR−c, DIR−e, DIR−f, DIR−g, DIR−h, and DIR−i [6,7,9]. The DIR−a subfamily members, AtDIR6 and pbDIR, have been reported to be involved in the process of lignin synthesis and show differential expression following abscisic acid, salicylic acid, and methyl jasmonate treatments, suggesting that DIR proteins can respond to exogenous attacks on plants [10,11]. However, soybean GmDRR1 proteins belonging to the DIR−b/d subfamily have been found to be involved in soybean resistance to *Phytophthora sojae*. In this study, 64 sugarcane sequences were divided into six subfamilies (Figure 1). Sugarcane sequences were distributed across all subfamilies, indicating that the functions of the ScDIR protein are not limited to inhibition of disease resistance and lignin formation [1,19].

Previous studies have shown that almost all members of the DIR protein family contain a DIR structural domain [7,12]. In sugarcane, Damaj identified 42 *ScDIR* genes; analysis by homology matching showed that each of these genes contained a DIR structural domain, presumably with similar gene functions [20,21]. Each of the 64 sugarcane ScDIR proteins in this study contained at least one DIR domain, and a few also included JAC and PK domains (Appendix A). The functions of ScDIR protein in sugarcane may be complex and diverse. Bioinformatics analysis of four drought−induced sugarcane ScDIR genes showed that the ScDIR5 and ScDIR11 proteins belonged to the DIR−b/d subfamily, whereas ScDIR7 and ScDIR40 belonged to the DIR−c subfamily (Figure 1), suggesting that these two subfamily members are involved in plant response to abiotic stress. We also found that these four ScDIR proteins had similar structures, with three common motifs appearing in the DIR functional domain (Figure 3).

To date, many studies have shown that DIR proteins participate in the synthesis of lignin and in the secondary metabolism of plants, thereby regulating the immune defense response of plants to external stress [22,23]. *SHDIR16* has been shown to be induced by salicylic acid, jasmonic acid, or methyl jasmonate in sugarcane [21], and a *ScDIR* gene that was specifically expressed in stems after abiotic stress was reported to enhance the tolerance of *Escherichia coli* to PEG and NaCl when expressed in the bacterium [18]. In this study, *ScDIR* expression levels (Figure 5) and lignin content (Figure 6) were significantly increased in transgenic tobacco plants, further confirming that the sugarcane *ScDIR* gene is involved in lignin synthesis and participates in plant resistance to drought stress.

Regarding drought stress in transgenic tobacco, we found that the *ScDIR11*−transgenic lines with the most significant expression did not have the highest drought tolerance (Table 1). The *ScDIR7*−transgenic line had the highest lignin content (Figure 6) and the strongest drought tolerance. Physiological and biochemical analyses also showed that the *ScDIR7*−transgenic line had the highest SOD and CAT activities (Figure 7b,c). However, the lignin content of the *ScDIR11*−transgenic line was the second−highest, and it had strong drought tolerance. The *ScDIR5* and *ScDIR40*−transgenic lines had poor drought tolerance, but their SOD and CAT activities differed (Figure 7b,c), indicating an inconsistent degree of internal damage. The results of this study are similar to those of Li et al., which indicated that the enzyme activity of plants was affected to varying degrees [24,25].

Previous studies have shown that the concentrations of H_2_O_2_ and MDA increased in plants after abiotic stress [26]. Our data also showed that H_2_O_2_ and MDA concentrations increased after transgenic tobacco plants were subjected to drought stress (Figure 7d,e). This indicated that the oxidized intracellular nucleic acids and membrane lipids of the plants were subjected to a high degree of peroxidation, resulting in cell damage and senescence [27,28]. After drought stress, the physiological and biochemical indexes of the *ScDIR5*−transgenic and *ScDIR40*−transgenic lines were the same as those of the *ScDIR7*−transgenic and *ScDIR11*−transgenic lines, but the drought resistance was weak, indicating that a single indicator cannot accurately determine plant tolerance.

In conclusion, this study found that the expression levels of four *ScDIR* genes in sugarcane were significantly increased after drought stress. These four genes were then overexpressed in tobacco, and the lignin content of the transgenic plants increased. In addition, the transgenic plants were significantly more resistant to drought stress. These results demonstrate that *ScDIR* genes enhance the stress resistance of sugarcane, providing a basis for further research on the role of *ScDIR* gene families in sugarcane defense, especially in the stage of cell wall defense.

## 4. Materials and Methods

### 4.1. Plant Materials and Drought Stress Conditions

The sugarcane variety, ROC22, was maintained in the Collaborative Innovation Center of the Sugarcane Industry at Guangxi University, in Fuisui county of Guangxi province, China. Sugarcane and tobacco seedlings were cultured in 10−cm pots with growing substrates (Nanning Guiyuxin Agricultural Science and Technology, Nanning, China) in a greenhouse at 28 °C for 5 days. Drought stress was induced by watering the seedlings with 10 mL of 20% polyethylene glycol (PEG) 6000 per pot every 2 days. After the treatment, the stems and leaves of sugarcane were harvested on days 0, 10, and 20; rapidly frozen in liquid nitrogen, and then stored at −80 °C for the extraction of the total RNA. Each sample consisted of three biological replicates. Tobacco (*Nicotiana benthamiana*) was grown in a growth chamber at 28 °C with 16 h light and 8 h dark.

### 4.2. Bioinformatics Analysis of Sugarcane ScDIR Proteins

The DIR protein sequences of sugarcane were obtained by searching the Sugarcane Genome Hub (Sugarcane Genome Hub (cirad.fr, accessed on 1 May 2022)), NCBI (National Center for Biotechnology Information (nih.gov, accessed on 1 May 2022)), CTBE genome (https://www.researchgate.net/publication/305766209_Surveying_the_complex_polyploid_sugarcane_genome_sequence_using_synthetic_long_reads, accessed on 1 May 2022, Brazilian Bioethanol Science and Technology Laboratory CTBE, Rua Giuseppe Maximo Scolfaro, Campinas, Brazil), and SUCEST databases (http://sucestfun.org, accessed on 1 May 2022), and DIR protein sequences from other plants were obtained from the RiceData (https://ricedata.cn/, accessed on 1 May 2022) and NCBI databases, following the method of [9]. DIR proteins were classified based on the maximum likelihood method of the modified Poisson model; evolutionary analysis was performed in MEGAX (https://www.megasoftware.net, accessed on 1 May 2022) [29], and protein sequence alignment was performed with DNAMAN software (https://www.lynnon.com/dnaman.html, accessed on 1 May 2022). Motif structures were analyzed using the MEME Suite (meme−suite.org, accessed on 1 May 2022). Plant−PLoc software was used to predict the gene expression locations in plants (Plant−PLoc server−SJTU). The tertiary structures of proteins were modeled with SWISS−MODEL online software (https://swissmodel.expasy.org/, accessed on 1 May 2022) using the pterocarpan synthase DIR protein (PsPTS1) as a template.

### 4.3. Gene Expression Quantification

The total RNA was extracted from the plants using TransZol Plant (Transgen, Beijing, China) according to the manufacturer’s protocol. First−strand cDNA was reverse transcribed using the HiScript^®®^ II 1st Strand cDNA Synthesis Kit (+gDNA wiper) (Vazyme, Nanjing, China). Quantitative real−time PCR (qRT−PCR) was performed using the PerfectStart^®®^ Green qPCR SuperMix(+Dye II) (Transgen, Beijing, China) in a Light Cycler 480 II fluorescent PCR instrument (Roche Diagnostics Corporation, Indianapolis, Indiana, USA). The qRT−PCR protocol was as follows: 95 °C for 5 min and 40 cycles of 95 °C for 30 s, 60 °C for 30 s, 72 °C for 1 min, followed by a final extension at 72 °C for 7 min. Using the *GAPDH* gene as an internal control, the relative expression levels in the transgenic tobacco lines and non−transgenic tobacco were detected. The 2^−^^∆∆Ct^ method was used for quantification [30], and the variation in expression was analyzed based on three biological replicates. The primers used for the qRT−PCR are shown in Appendix A.

### 4.4. Subcellular Localization of Proteins

The coding region of each target protein gene was cloned into the pCAMBIA3300−GFP vector. The recombinant constructs were transferred into *Agrobacterium* EHA105 using standard cloning techniques. Chopped onion inner scales (1 cm × 1 cm) were soaked in 6 mL of *Agrobacterium* resuspended in MS (Murashige and Skoog Basal Salt Mixture, 30 g·L^−1^ of sucrose and 0.7% (g·v^−1^) agar, pH 5.7) medium containing 10 mmol·L^−1^ of MgCl_2_ and 120 μmol·L^−1^ of AS, at an optical density of 1.0 (wavelength 600 nm) for 30 min at 28 °C. The onion scales were then transferred to Petri dishes containing 25 mL of half−concentration MS and co−cultured with *Agrobacterium* for 1 day [31]. The onion’s inner epidermis was separated in 0.3 g·mL^−1^ of sucrose solution and then transferred to a glass slide. The GFP fluorescence was observed using an Olympus BX41 microscope (Olympus Corporation, Olympus, Japan).

### 4.5. Tobacco Transformation and Regeneration

The coding region of each target gene was cloned into a binary vector, pCAMBIA3300, under the control of the CaMV 35S promoter. The resulting constructs were introduced into tobacco callus using *Agrobacterium*−mediated transformation [32]. Seeds from the transformed plants (T0) were harvested and sowed on an MS medium containing 0.1 mg·L^−1^ of glufosinate. The T1 and T2 transformants were screened and verified by PCR. Positive T2 transformants were used for further analyses.

### 4.6. Measurement of Physiological and Biochemical Indices

The comparisons of physiological and biochemical indices between wild−type tobacco and transgenic tobacco under normal conditions and abiotic stresses were performed. Four physiological and biological indices (SOD activity, MDA content, CAT activity, and MDA content) were measured from tobacco leaves according to the instructions supplied with the kit (Solarbio, Beijing, China). The results and standard errors were calculated based on data obtained from three independent experiments. Significant differences were determined by *t*−test.

### 4.7. Statistical Analyses

Principal component analysis (PCA) was performed, and subordinate functions were calculated using the method of [33]. The drought resistance coefficient (DRC) was calculated as follows: RC (%) = measured value in the treatment area/measured value in the control area × 100% [34]. SPSS (version 26.0) was used for statistical analysis.

## Figures and Tables

**Figure 1 ijms-23-05340-f001:**
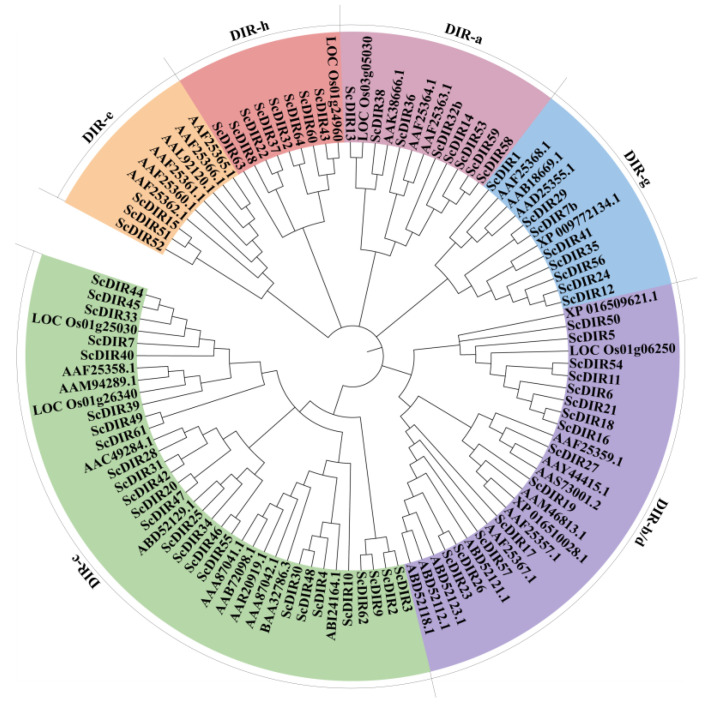
Classification of 64 sugarcane ScDIR proteins and classical DIR proteins from 12 other plants. Using MEGAX software, a phylogenetic tree was constructed based on the full−length protein sequences of 64 sugarcane ScDIR proteins and 12 ScDIRs from other species. The analysis was performed using Maximum Likelihood under the evolutionary model with 500 bootstrap replicates. The six subfamilies are indicated in different colors. The sequence names and accession numbers for each clade are shown in Appendix A.

**Figure 2 ijms-23-05340-f002:**
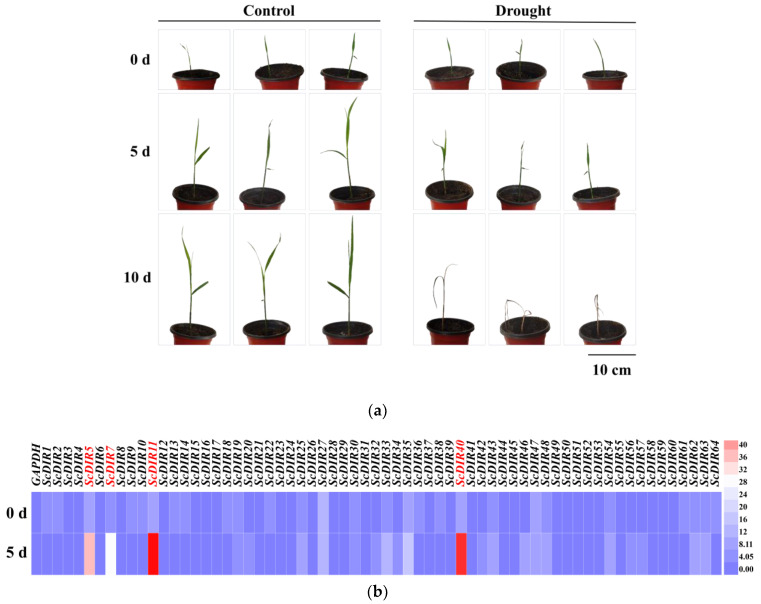
Relative expression pattern of sugarcane *ScDIR* genes under drought stress conditions. (**a**) Phenotypes of sugarcane seedlings at 0, 5, and 10 days of drought stress. (**b**) Relative expression of 64 *ScDIR* genes as measured by qRT−PCR. Analysis of expression differences in sugarcane seedlings after 0 and 5 days of drought stress compared with those in the absence of stress. Heatmap generation and hierarchical clustering were performed using the Heml 1.0.3.7 software package. The color scale below the heat map indicates expression values; blue indicates low transcript abundance while red indicates a high level of transcript abundance. *GAPDH* was used as an internal reference gene.

**Figure 3 ijms-23-05340-f003:**
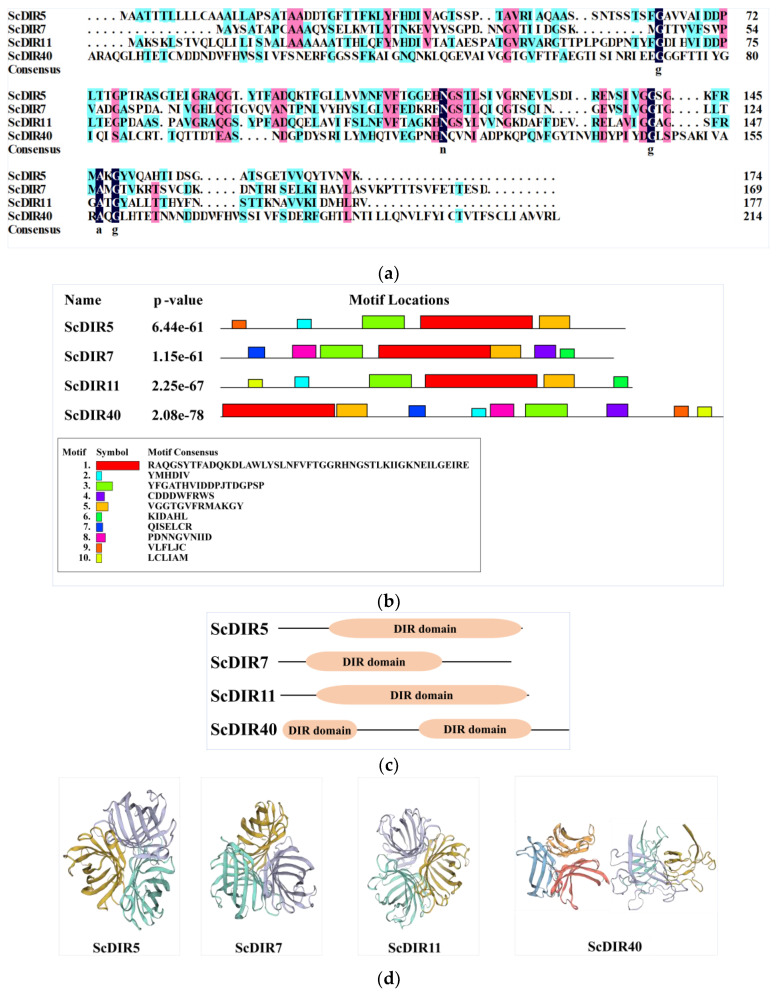
Protein sequence analysis, conserved domains, and tertiary structures of ScDIR5, ScDIR7, ScDIR11, and ScDIR40. (**a**) protein sequence alignment. (**b**) conserved protein motifs predicted with MEME Suite software. (**c**) DIR domain. (**d**) tertiary structures of the ScDIRs predicted using SWISS−MODEL online software, using PsPTS1 as a template. Two structures were predicted in ScDIR40, one of them (right−most) may not be functional.

**Figure 4 ijms-23-05340-f004:**
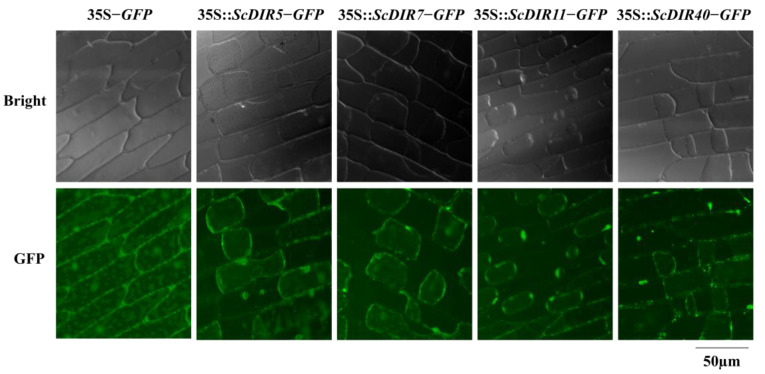
Transient expression of GFP fusion protein in onion epidermal cells. All constructs were driven by a 35S promoter. *ScDIR−GFP*, *GFP* fused with *ScDIR* at the C terminus.

**Figure 5 ijms-23-05340-f005:**
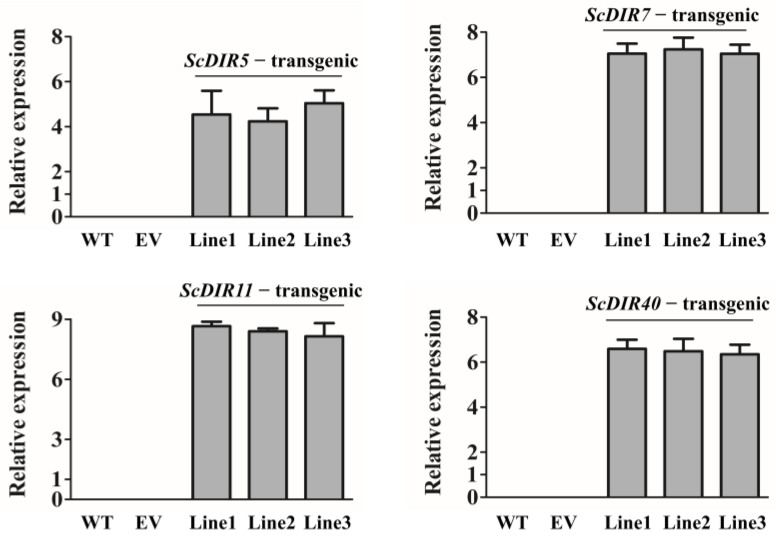
qRT−PCR analysis of *ScDIR5*, *ScDIR7*, *ScDIR11*, and *ScDIR40* transcript levels in three independent lines, respectively. A, Expression levels of the *ScDIR5* gene. B, Expression levels of the *ScDIR7* gene. C, Expression levels of the *ScDIR11* gene. D, Expression levels of the *ScDIR40* gene. Relative expression levels of *ScDIR* genes were determined by qRT−PCR analysis in wild−type tobacco (WT), tobacco with empty vector (EV), and transgenic tobacco (*ScDIR5*−transgenic, *ScDIR7*−transgenic, *ScDIR11*−transgenic, and *ScDIR40*−transgenic lines). The *GAPDH* gene was used as an internal control. Error bars denote the standard deviation calculated from three independent experiments.

**Figure 6 ijms-23-05340-f006:**
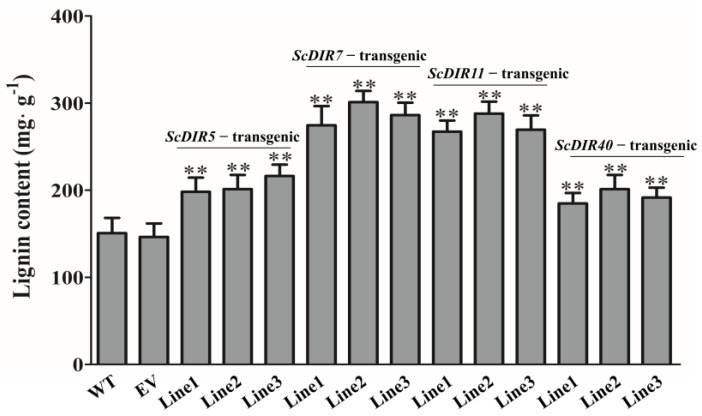
Lignin contents of *ScDIR* transgenic lines. Lignin content of tobacco leaves at the five−leaf stage was determined using a lignin assay kit. Error bars denote the standard deviation calculated from three independent experiments. Significant differences were determined by *t*−test. ** *p* < 0.01.

**Figure 7 ijms-23-05340-f007:**
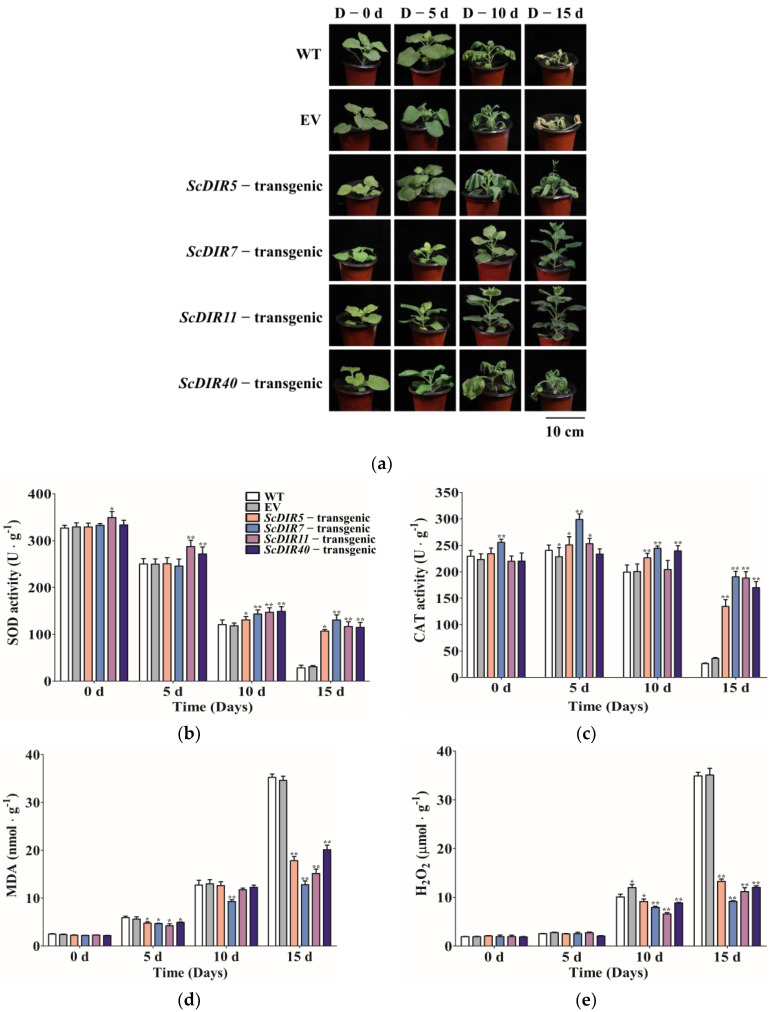
Phenotypes and physiological indices of *ScDIR* transgenic lines after drought stress. (**a**), Drought tolerance of transgenic tobacco plants overexpressing *ScDIR* genes. Photographs were taken before and after drought treatment. (**b**–**e**), Indexes of SOD, CAT, MDA, and H_2_O_2_ in WT and *ScDIR* transgenic lines. Error bars denote the standard deviation calculated from three independent experiments. Significant differences were determined by *t*−test. * *p* < 0.05, ** *p* < 0.01.

**Table 1 ijms-23-05340-t001:** Subordinate function values of principal components and comprehensive evaluation of drought tolerance in transgenic *n. benthamiana*.

Variety	Subordinate Function Values *		Integrated Assessment	Order of Resistance
	U1	U2	Value D	
Wild type	0.932	0.194	0.868	5
EV	1.024	0.191	0.834	6
*ScDIR5*−transgenic	1.258	0.228	1.178	4
*ScDIR7*−transgenic	1.689	0.277	1.587	1
*ScDIR11*−transgenic	1.521	0.261	1.428	2
*ScDIR40*−transgenic	1.272	0.198	1.202	3

* U1 and U2 indicate the subordinate function value of principal components 1 and 2, respectively.

## Data Availability

Not applicable.

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
