# Peer review of "Overexpression of Sugarcane ScDIR Genes Enhances Drought Tolerance in Nicotiana benthamiana"

_ijms, 2022, doi:10.3390/ijms23105340_

Round 1
Reviewer 1 Report
The authors characterize the dirigent genes from sugarcane. They identify 64 gene variants from the sugarcane database and use qt-PCR to determine expression levels of these genes after exposure to drought in sugarcane seedlings. Four genes ScDIR5, ScDir7, ScDir11 and ScDIR 40, which show increased expression after drought stress are examined further. Transient expression and cellular localization are attempted using onion protoplasts. Constitutive overexpression in tobacco, followed by plant responses to drought stress were also determined. The manuscript successfully characterizes dirigent genes in sugarcane and does demonstrate response of some of the genes to drought stress.
The onion transient expression and subcellular localization experiments are questionable, do not add any significant data to the manuscript, and should be removed. Localization of the genes was determined after 24 hrs after cocultivation with Agrobacterium. A more robust approach would have been to transform tobacco with the GFP vector, generate stable transformant, and then determine localization. The GFP construct needs better definition. For processing of the membrane and extracellular dirigent proteins was a signal sequence used, if so, where is the GFP fusion?
Table 2b can be strengthened by adding the names ScDIR5, ScDIr7, ScDIR11 and ScDIR40 to the heat map.
The ScDIR genes in this manuscript should be added to the Genbank database and given accession numbers as a condition of publication.
Author Response
Dear Sir/Madam:
First of all, thank you very much for your questions and suggestions. The following are responses to your comments and suggestions.
- The onion transient expression and subcellular localization experiments are questionable, do not add any significant data to the manuscript, and should be removed. Localization of the genes was determined after 24 hrs after cocultivation with Agrobacterium. A more robust approach would have been to transform tobacco with the GFP vector, generate stable transformant, and then determine localization. The GFP construct needs better definition. For processing of the membrane and extracellular dirigent proteins was a signal sequence used, if so, where is the GFP fusion?
Reply: The onion transient expression and subcellular localization experiments refer to this article: Identifying Subcellular Protein Localization with Fluorescent Protein Fusions After Transient Expression in Onion Epidermal Cells. Your proposal to genetically transform a GFP-containing vector into tobacco for subcellular localization will be considered in the future. Dirigent proteins are mainly stable proteins, so the location of the DIR-GFP fusion protein is the location of the protein.
- Table 2b can be strengthened by adding the names ScDIR5, ScDIr7, ScDIR11 and ScDIR40 to the heat map.
Reply: Due to upload issues, we have re-labeled ScDIR in the heatmap in figure 2b and accepted your suggestion to highlight ScDIR5, ScDIR7, ScDIR11, and ScDIR40.
- The ScDIR genes in this manuscript should be added to the Genbank database and given accession numbers as a condition of publication.
Reply: The 64 ScDIR genes mentioned are from multiple databases such as CTBE genome, RNAseq, and SUCEST, and their accession numbers and databases are listed in Table S1.
This information has also been incorporated into the text in the revision.
Reviewer 2 Report
In my opinion, the manuscript submitted for review is of a good scientific standard. I like the very idea of the manuscript, the research and the descriptive part, however, the graphic part needs to be refined - the figures are of poor quality. I do not know if it is the fault of 1) the conversion from ms word to PDF? 2) or the poor quality of the original photos. I ask the authors to improve these parts of the manuscript.
Author Response
Dear Sir/Madam:
First of all, thank you very much for your questions and suggestions. The following are responses to your comments and suggestions.
- In my opinion, the manuscript submitted for review is of a good scientific standard.I like the very idea of the manuscript, the research and the descriptive part, however, the graphic part needs to be refined - the figures are of poor quality. I do not know if it is the fault of 1) the conversion from ms word to PDF? 2) or the poor quality of the original photos. I ask the authors to improve these parts of the manuscript.
Reply: The images in the manuscript are compressed by word, and I will resubmit the images in clear resolution.
This information has also been incorporated into the text in the revision.